# Active invasion of bacteria into living fungal cells

**Nadine Moebius[1†], Zerrin Üzüm[1†], Jan Dijksterhuis[2], Gerald Lackner[1], Christian Hertweck[1,3]\***

[1]Department of Biomolecular Chemistry, Leibniz Institute for Natural Product Research and Infection Biology, Jena, Germany; [2]CBS-KNAW Fungal Biodiversity Centre, Utrecht, Netherlands; [3]Friedrich Schiller University, Jena, Germany

**Abstract** The rice seedling blight fungus *Rhizopus microsporus* and its endosymbiont *Burkholderia rhizoxinica* form an unusual, highly specific alliance to produce the highly potent antimitotic phytotoxin rhizoxin. Yet, it has remained a riddle how bacteria invade the fungal cells. Genome mining for potential symbiosis factors and functional analyses revealed that a type 2 secretion system (T2SS) of the bacterial endosymbiont is required for the formation of the endosymbiosis. Comparative proteome analyses show that the T2SS releases chitinolytic enzymes (chitinase, chitosanase) and chitin-binding proteins. The genes responsible for chitinolytic proteins and T2SS components are highly expressed during infection. Through targeted gene knock-outs, sporulation assays and microscopic investigations we found that chitinase is essential for bacteria to enter hyphae. Unprecedented snapshots of the traceless bacterial intrusion were obtained using cryo-electron microscopy. Beyond unveiling the pivotal role of chitinolytic enzymes in the active invasion of a fungus by bacteria, these findings grant unprecedented insight into the fungal cell wall penetration and symbiosis formation.

**\*For correspondence:** christian.
hertweck@hki-jena.de

[†]These authors contributed
equally to this work

**Competing interests:** The
authors declare that no
competing interests exist.

**Reviewing editor:** Thorsten
Nürnberger, University of
Tübingen, Germany

## Introduction

Interactions between bacteria and fungi are widespread in nature and play pivotal roles in ecological and medicinal processes (*Frey-Klett et al., 2011*). Moreover, fungal-bacterial associations are widely used for the preservation of the environment (e.g., mycorrhizae in reforestation), agriculture (e.g., food processing), and biotechnology (e.g., pharmaceutical research) (*Scherlach et al., 2013*). Beyond the most commonly observed microbial cell–cell interactions, there is a growing number of known endosymbioses where bacteria dwell within fungal hyphae (*Bonfante and Anca, 2009*; *Kobayashi and Crouch, 2009*; *Lackner et al., 2009b*; *Frey-Klett et al., 2011*). Symbioses with endofungal bacteria are often overlooked, yet they may have a profound effect on the host's lifestyle. Bacterial endosymbionts of arbuscular mycorrhizal fungi, for example, might be implicated in the vitamin B12 provision for the fungus (*Ghignone et al., 2012*). Endobacteria *Rhizobium radiobacter*, isolated from the mycorrhizal fungus, exhibit the same growth promoting effects and induce systemic resistance to plant pathogenic fungi in the same way that the fungus harboring the endobacteria does. Thus, it was proposed that the beneficial effects for the plant result directly from the presence of bacteria (*Sharma et al., 2008*). The rice seedling blight fungus, *Rhizopus microsporus*, and its endosymbiont bacterium, *Burkholderia rhizoxinica* represent a particularly noteworthy example of a bacterial-fungal endosymbiosis (*Partida-Martinez and Hertweck, 2005*; *Lackner and Hertweck, 2011*). The fungus harbors endosymbionts of the genus *Burkholderia*, which reside within the fungal cytosol, as shown by confocal laser scanning microscopy, transmission electron microscopy (EM) and freeze–fracture EM (*Partida-Martinez et al., 2007a*, *2007b*, *2007c*). The bacteria are harnessed by the fungus as producers of highly potent antimitotic macrolides (*Scherlach et al., 2006*), which are then further processed

**eLife digest** Many organisms live in what are known as symbiotic relationships, whereby two or more different species share a close and often long lasting association. Examples of symbiosis abound in nature, with well-studied examples including the sea anemone and the clownfish, corals and their photosynthetic algae, and nitrogen-fixing bacteria that live in association with legumes.

In some cases, each member of the relationship benefits from the symbiosis. One such example is the rice seedling blight fungus *Rhizopus microsporus* and its bacterial symbiont *Burkholderia rhizoxinica*, which work together to produce toxins that kill rice plants. This frees up nutrients that nourish both the fungus and the bacteria. *B. rhizoxinica* lives inside the tissues of the fungus, but to do so the bacterial cells must first travel through the tough cell walls of the fungus. How these bacteria do this without also damaging the fungus was unknown.

Moebius et al. discovered that *B. rhizoxinica* gains access into *R. microsporus* cells by using a series of proteins in the bacterium's membrane called the type 2 secretion system, which transport proteins from the inside of the cell to the outside. On analyzing the proteins released by this system, Moebius et al. identified several enzymes that help the bacteria attach to the fungal cell wall and soften it so that the bacteria can penetrate into the cell. Only small amounts of enzyme are needed for the softening process, meaning that penetrating the cell wall is a relatively gentle process that causes no lasting damage to the fungus. Moebius et al. also captured images of the bacteria invading the fungal cells using a technique called cryo-electron microscopy, and provide the first known images of this type of infection in progress.

by the host into the phytotoxin rhizoxin (*Scherlach et al., 2012*). The toxin represents the causative agent of rice seedling blight, which weakens or kills the rice plants (*Lackner et al., 2009b*). Both the saprotrophic fungus and the endofungal bacteria benefit from the nutrients released, and *R. microsporus* provides a protective shelter for the bacterial partner. The *Rhizopus-Burkholderia* association also stands out as it employs an elegant mechanism that allows the persistence and spreading of the symbiosis through spores containing the endosymbionts (*Partida-Martinez et al., 2007c*) (*Figure 1*). Yet it is unknown how the vegetative reproduction of the fungus has become totally dependent upon the presence of the endobacteria (*Partida-Martinez et al., 2007c*). Insights into the genome of *B. rhizoxinica* and mutational studies have unveiled several symbiosis factors (*Leone et al., 2010*; *Lackner et al., 2011a*, *2011b*).

A plausible scenario for the evolution of the symbiosis is a shift from antibiosis or antagonism to mutualism. The rhizoxin complex secreted by the bacteria arrests mitosis in almost all eukaryotic cells. Yet, *Rhizopus*, amongst other zygomycetes, has gained resistance to this toxin due to a mutation at the β-tubulin binding site (*Schmitt et al., 2008*). Furthermore, phylogenetic analyses point to host switching events during evolution (*Lackner et al., 2009a*), which is also supported by the engagement of an *hrp* locus of *B. rhizoxinica* (*Lackner et al., 2011a*). In addition to this, the LPS layer of the *B. rhizoxinica* is known to be unique to its niche, due to high resemblance to fungal sugar content (*Leone et al., 2010*). Although there is ample knowledge on the persistence of the symbiosis, it has remained fully enigmatic how the bacteria enter the fungal cells. Interestingly, there is no sign of endo-/phagocytosis, which rules out a major avenue of bacterial colonization (*Partida-Martinez and Hertweck, 2005*; *Partida-Martinez et al., 2007a*, *2007c*).

Bacterial invasion of eukaryotic cells is a major area of research in infection biology, and a large body of knowledge has been gathered on the pathogen's strategies to invade host cells (*Cossart and Sansonetti, 2004*). In addition to host driven endocytosis, a number of enzymes have been described that act locally to damage host cells and to facilitate the entry of the pathogen into the tissue (*Harrison, 1999*; *King et al., 2003*). Yet, this knowledge is limited to the invasion of human, animal and plant cells. It has been reported that some bacteria employ extracellular enzymes for mycophagy (*Leveau and Preston, 2008*). However, despite a growing number of described fungal endobacteria (*Lackner et al., 2009b*; *Frey-Klett et al., 2011*), there is a striking lack of knowledge about the avenues and active mechanisms that permit fusion with or entry into fungal hyphae, where the fungus is left intact to serve as a host for the endobacteria. Here we report the genomics- and proteomics-driven discovery of a new bacterial invasion process that involves the secretion of

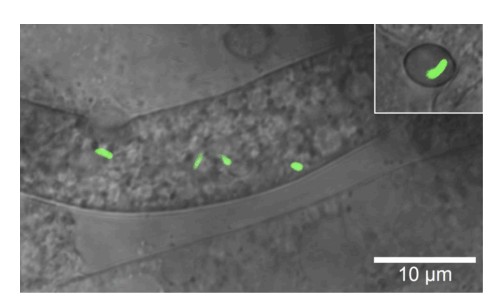

**Figure 1**. Microscopic image of *Burkholderia rhizoxinica* (green) residing in the cytosol of *Rhizopus microsporus*. The GFP encoding *B. rhizoxinica* cells can re-colonize the sterile *R. microsporus*, then induce fungal sporulation. The endobacterium is transmitted via fungal vegetative spores (upper right corner).

chitinolytic enzymes. Furthermore, we present the first electron microscopic snapshots of the actual infection process.

## Results

### A type 2 secretion system (T2SS) of the *Burkholderia* endosymbiont is essential for the formation of the endosymbiosis

Both pathogens (or antagonists) and mutualists often employ the same mechanisms during the infection process (*Dale and Moran, 2006*). Thus, we mined the gene repertoire of *B. rhizoxinica* (*Lackner, 2011b*) for potential molecular infection mechanisms known from pathogenic bacteria. A type 2 secretion system (T2SS), also called general secretion pathway (gsp), encoded by a 12 kb gene cluster on the *B. rhizoxinica* chromosome (*Table 1*) seemed to be a promising candidate to enable the bacterium to enter the host. T2SS are typically involved in the secretion of various toxins and lytic enzymes (*Cianciotto, 2005*; *Korotkov et al., 2012*) and the overall organization of T2SS gene clusters is well conserved between related species (*Figure 2*). We generated targeted deletion mutants to investigate the role of the T2SS in the infection process. Specifically, we selected *gspC* and *gspD*, since their gene products are essential proteins of the type 2 secretion machinery in related bacteria (*DeShazer et al., 1999*; *Korotkov et al., 2011*). The outer membrane pore is formed presumably by the multimeric secretin GspD, and GspC appears to link the inner and outer membranes by providing the contact to GspD via a homology region (*Korotkov et al., 2011*). Although it is notoriously difficult to genetically modify the symbiotic bacteria, we succeeded in generating ΔgspC and ΔgspD mutants using a double-crossover strategy.

To begin, we addressed the proteolytic potential of *B. rhizoxinica* wild type (wt) and the T2SS defective mutants to evaluate the effect of the knock-outs (*Figure 3*, *Figure 3—figure supplement 1*). Using a skim milk plate assay we detected strong proteolytic activity in the wt supernatant, while the T2SS mutants showed no activity (*Figure 3—figure supplement 3*). The ability of the isolated endobacteria to re-infect the fungus and to control fungal sporulation was examined using a sporulation bioassay. The appearance of mature sporangia that form sporangiospores is seen as an indication of a successful establishment of the symbiosis. In co-cultures of wt *B. rhizoxinica* and the cured fungal host, sporulation is visible after 2–3 days. In contrast, there was absolutely no visible spore formation upon co-cultivation with *B. rhizoxinica* ΔgspD::Kan^r or ΔgspC::Kan^r (*Figure 3*). Furthermore, fluorescence microscopy proved to be most helpful to distinguish between mutants defective in colonization or induction of fungal sporulation. A constitutive GFP-expressing strain allowed monitoring of the invasion of bacteria into the fungal hyphae. While fluorescent bacteria with an intact T2SS were able to enter the fungal cells, there was no detection of any endobacteria when either of the T2SS mutants was co-cultured with the cured fungus (*Figure 3*, *Figure 3—figure supplement 2*).

### The T2SS releases chitinolytic and chitin-binding proteins

In order to identify the secreted factors that could play a role in the bacterial fungal interaction we performed comparative 2-D gel electrophoresis of the exoproteomes (secretomes) of wt and mutant bacteria (*Figure 4*). The secretome analysis of the T2SS mutants showed a substantial reduction in the total protein yield (0.1% ± 0–015% of the wt secretome) despite an intense protein precipitation with fivefold TCA and the requirement of 100 µg of protein from each samples to be loaded on 2-D gel electrophoresis. In the wt secretome we identified surprisingly few proteins, although they were present in great abundance. Using MALDI-TOF we were able to detect the majority of the secreted proteins as chitin-binding protein (Cbp) and chitosanase (Chts), which are encoded in the bacterial genome. The chitin-binding protein belongs to the non-catalytic carbohydrate-binding proteins of the CBM33 family (*Henrissat and Davies, 2000*), which can bind to chitin and facilitate the action of

**Table 1.** Annotation of the *B. rhizoxinica* T2SS gene cluster

| Gene | Proposed function of encoded protein | Homolog in *Klebsiella oxytoca* | Percent identity (BlastP) | Homolog in *Burkholderia pseudomallei* | Percent identity (BlastP) |
|---|---|---|---|---|---|
| gspC | Connecting inner and outer membrane complex | pulC | – | gspC | 61 |
| gspD | Secretin, outer membrane pore formation | pulD | 39 | gspD | 64 |
| gspE | Cytoplasmitc ATPase, energy for translocation of pseudopilins | pulE | 54 | gspE | 89 |
| gspF | Anchoring protein, inner membrane platform for pseudopilins | pulF | 46 | gspF | 83 |
| gspG | Major prepilin-like protein, pilus-like structure formation | pulG | 56 | gspG | 87 |
| gspH | Pseudopilin subunit, pilus-like structure formation | pulH | 70 | gspH | 66 |
| gspI | Pseudopilin subunit, pilus-like structure formation | pulI | 44 | gspI | 66 |
| gspJ | Pseudopilin subunit, pilus-like structure formation | pulJ | 43 | gspJ | 55 |
| gspK | Pseudopilin subunit, pilus-like structure formation | pulK | 26 | gspK | 62 |
| gspL | Anchoring protein, inner membrane platform for pseudopilins | pulL | 27 | gspL | 57 |
| gspM | Anchoring protein, inner membrane platform for pseudopilins | pulM | 24 | gspM | 56 |
| gspN | Connecting inner and outer membrane complex | pulN | 29 | gspN | 63 |
| gspO | Prepilin, inner membrane peptidase | pulO | 43 | gspO | 60 |

chitinases. The chitosanase is part of the glycoside hydrolase family 46 with a specific hydrolytic activity on chitosan (***Henrissat and Davies, 2000***).

Chitin and chitosan are well known as major structural components of the fungal cell wall (***Gooday, 1990***; ***Adams, 2004***). Chitosan is a dominant component of the Zygomycete cell wall, but chitin is also abundant, as we could show by calcofluor staining of the *R. microsporus* cell wall (***Figure 5B***). While screening the *B. rhizoxinica* genome for genes for chitinolytic enzymes, we also detected a gene for

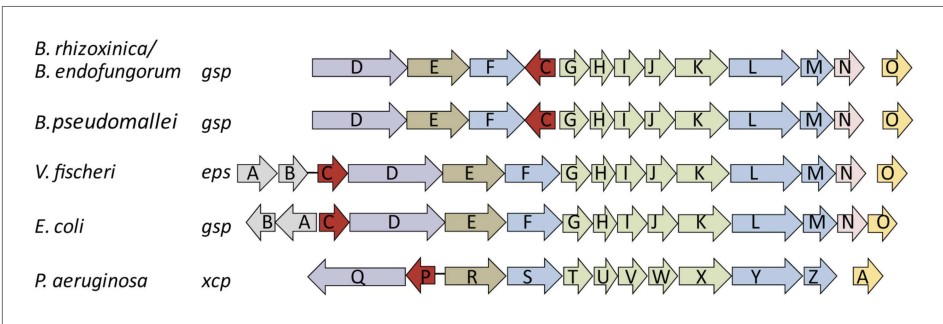

**Figure 2**. Schematic view of the organization of the type 2 secretion system (T2SS) gene clusters from various bacterial species. The T2SS gene loci of the two *R. microsporus* endosymbiotic bacterial strains *B. rhizoxinica* and *Burkholderia endofungorum*, as well as the squid endosymbiont *Vibrio fischeri*, *Escherichia coli*, and the human pathogens *Burkholderia pseudomallei* and *Pseudomonas aeruginosa* are displayed here. The spaces between the arrows represent non-adjacent genes (single genes are located further away on the genome), while lines indicate closely linked genes.

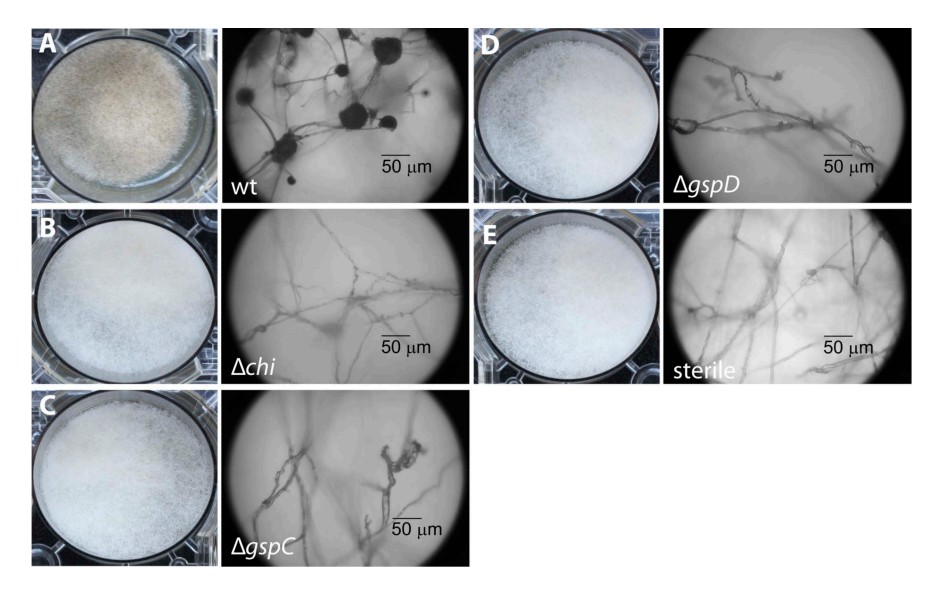

**Figure 3**. Photographs and microscopy images of *R. microsporus* hyphae several days after inoculation with *B. rhizoxinica* wt or mutant strains in 6-well plates. The pictures present the infection of *R. microsporus* with wt or mutant *B. rhizoxinica* in the following order: (**A**) *B. rhizoxinica* wt, (**B**) *B. rhizoxinica* Δ*chi*::Kan^r, (**C**) *B. rhizoxinica* Δ*gspC*::Kan^r, (**D**)—*B. rhizoxinica* Δ*gspD*::Kan^r, (**E**)—control (no bacteria added). Spore formation is visible in (**A**), while spore formation was not detected in (**B**–**E**) even after 5 days of co-incubation.

The following figure supplements are available for figure 3:

**Figure supplement 1**. Knock out strategy of site directed mutagenesis to *B. rhizoxinica*.

**Figure supplement 2**. Fluorescence microscopy carried out with *B. rhizoxinica* wt and mutant strains *B. rhizoxinica* Δ*gspD*, *B. rhizoxinica* Δ*gspC* and *B. rhizoxinica* Δ*chi* in a 3 day co-culture with sterile *R. microsporus*.

**Figure supplement 3**. Lytic potential of *B. rhizoxinica* secretome.

a chitinase that contains a signal sequence for secretion. The corresponding gene product could also in fact be detected by 2-D gel electrophoresis and MALDI analysis, albeit in lower abundance than Cbp and Chts. From the structure-based alignment and phylogenetic information (**Figure 5C**) we can conclude that the *B. rhizoxinica* chitinase (Chi) belongs to the family 18 chitinases in subfamily B. The closest structural homolog is PF-ChiA from *Pyrococcus furiosus* that has endochitinase activity (**Nakamura et al., 2007**).

A chitin-binding assay verified that all three proteins bind to chitin (**Figure 5D**). This finding is fully in line with the observation that the amount of chitinolytic proteins in the 2-D gel is greatly reduced when samples of fungal-bacterial co-cultures are applied (**Figure 4D**), most likely because the proteins bind to the fungal cell wall. Despite the total decrease in secreted protein amount, analysis of the T2SS mutants, Δ*gspD*::Kan^r and Δ*gspC*::Kan^r, revealed that all three chitinolytic proteins are substantially reduced in the secretome. Chitinase could not be detected in the secretome of Δ*gspD*::Kan^r, even when a 10-fold amount of precipitated secretome was loaded on the gel (**Figure 4C**). This indicates that the detected proteins are all collected from the dead cells rather than being excreted, and proving the selective secretion of chitinolytic proteins through T2SS.

## Chitinase is crucial for bacteria to enter hyphae

In order to investigate which chitinolytic proteins are essential for hydrolyzing the fungal cell wall we individually deleted the corresponding genes. Using the gene deletion strategy described above, we successfully obtained the mutants Δ*chi*::Kan^r, Δ*cbp*::Kan^r and Δ*chts*::Kan^r. All three mutants were tested in the previously described sporulation assay. The chitosanase and the chitin-binding protein

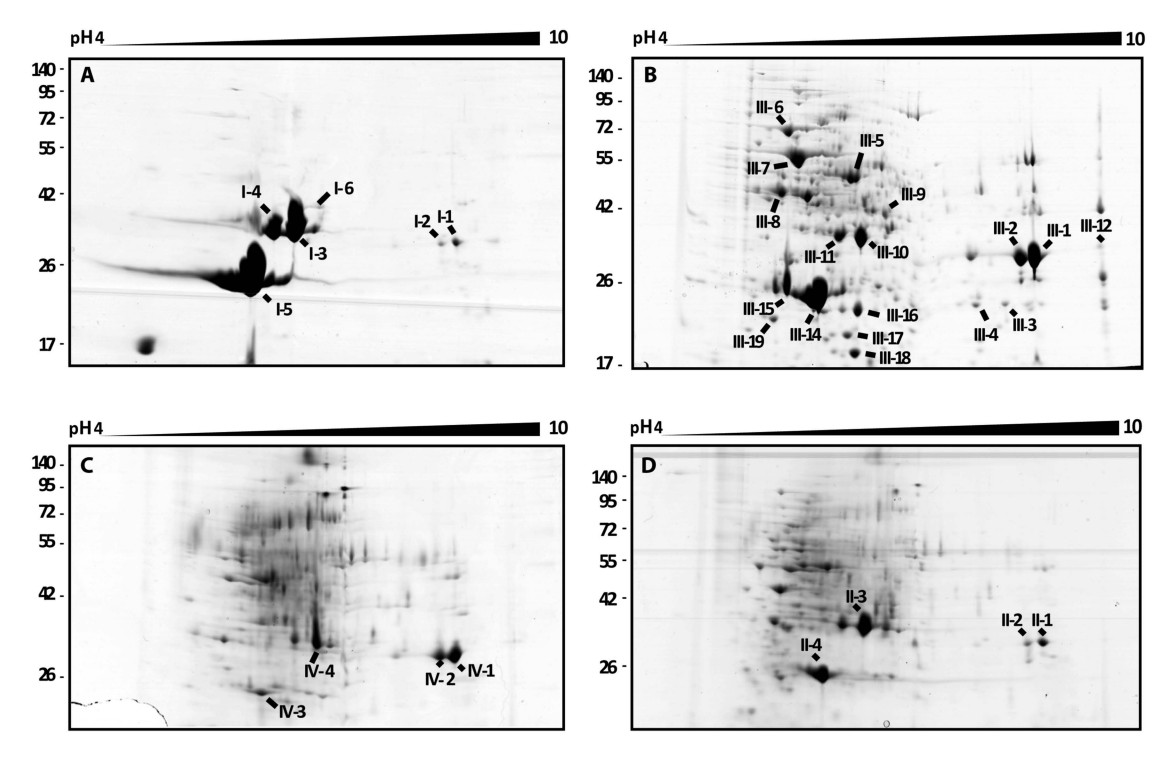

**Figure 4**. 2D gel analysis of the secretomes of wild type and mutants. (**A**) *B. rhizoxinica* wt, (**B**) *B. rhizoxinica* Δ*gspC*::Kan^r, (**C**) *B. rhizoxinica* Δ*gspD*::Kanr and (**D**) *B. rhizoxinica* wt in co-culture with *R. microsporus*. Chitin-binding protein (I-5, III-14, III-15, IV-6, II-5), chitinase (I-6, III-9) and chitosanase (I-3, I-4, III-10, III-11, IV-3, IV-4, IV-5, II-3) were identified. List of identified proteins is shown in *Table 2*. In general, each gel was loaded with 100 μg of TCA-precipitated proteins from the culture supernatant.

null mutants restored the symbiosis and retained their ability to illicit spore formation. In contrast, the chitinase deletion strain was no longer able to induce spore formation even after 1 week of extended co-culture (*Figure 3*). By using fluorescence microscopy, we found that bacteria that were incapable of producing chitinase could not invade fungal cells.

The culture supernatant of a pure wild-type *B. rhizoxinica* culture showed weak activity in a chitinolytic plate assay. Also no significant activity within the *B. rhizoxinica* secretomes could be detected in an assay with the aqueous substrate CM-chitin-RBV. To unequivocally prove its ability for chitinolysis, the *chi* gene was cloned and chitinase was heterologously produced in *Escherichia coli*. The *B. rhizoxinica* chitinase-enriched secretome was tested in an assay with the aqueous substrate CM-chitin-RBV. A high chitinolytic activity could thus be observed that remained stable over several hours, while the *E. coli* expression host (negative control) showed no activity (*Figure 5A*).

## Genes for chitinolytic proteins and components of T2SS are highly expressed during infection

Next, we wanted to address the question whether or not the production of chitinolytic proteins is constitutive or dependent upon the presence of the fungal host. Therefore, we monitored the expression of genes coding for chitinolytic proteins (*chi, cbp* and *chts*) and components of the T2SS (*gspD* and *gspC*). All genes are 30- to 160-fold up-regulated in bacterial-fungal co-cultures compared to the pure bacterial culture (*Figure 5E*). Surprisingly, the expression levels after re-infection nearly decrease to the level found in pure culture. These results strongly indicate that all of the tested genes play a crucial role during the infection process although they are not required for the maintenance of the symbiosis.

## Bacteria caught in the act of infection using cryo-electron microscopy

According to our functional analyses, bacteria produce and secrete chitinolytic enzymes during infection. We reasoned that the bacteria employ these enzymes to locally digest the fungal cell wall,

**Table 2.** Secretome proteins identified by MALDI-TOF

| Spot no. | Accesion no. (NCBI) | Protein indetification | Total Mass0 (kDA) | Total pI | Mascot score | Matching peptides | Sequence coverage (%) | SignalP prediction (y/n) |
|---|---|---|---|---|---|---|---|---|
| I-1 | gi│312169059 | Glutamate/aspartate-binding protein | 32.4 | 9.6 | 64.2 | 13 | 40.1 | y |
| I-2 | gi│312169059 | Glutamate/aspartate-binding protein | 32.4 | 9.6 | 114.0 | 18 | 49.5 | y |
| I-3 | gi│312167773 | Chitosanase (EC 3.2.1.132) | 39.6 | 6.5 | 88.3 | 10 | 38.9 | y |
| I-4 | gi│312167773 | Chitosanase (EC 3.2.1.132) | 39.6 | 6.5 | 124.0 | 15 | 47.7 | y |
| I-5 | gi│312168534 | Chitin-binding protein | 27.0 | 7.7 | 85.2 | 8 | 49.2 | y |
| I-6 | gi│312168091 | Chitinase (EC 3.2.1.14) | 42.1 | 8.7 | 30.8 | 6 | 23.1 | y |
| II-1 | gi│312169059 | Glutamate/aspartate-binding protein | 32.4 | 9.6 | 107.0 | 20 | 51.5 | y |
| II-2 | gi│312169059 | Glutamate/aspartate-binding protein | 32.4 | 9.6 | 114.0 | 18 | 46.8 | y |
| II-3 | gi│312167773 | Chitosanase (EC 3.2.1.132) | 39.6 | 6.5 | 169.0 | 15 | 60.3 | y |
| II-4 | gi│312168534 | Chitin-binding protein | 27.0 | 7.7 | 80.4 | 7 | 29.1 | y |
| III-1 | gi│312169059 | Glutamate/aspartate-binding protein | 32.4 | 9.6 | 133.0 | 20 | 51.9 | y |
| III-2 | gi│312169059 | Glutamate/aspartate-binding protein | 32.4 | 9.6 | 96.2 | 19 | 51.2 | y |
| III-3 | gi│312167620 | Toluene transport system Ttg2d protein | 23.7 | 9.5 | 69.0 | 5 | 30.5 | y |
| III-4 | gi│312168022 | Ribosome recycling factor (RRF) | 21.0 | 9.0 | 74.0 | 9 | 44.1 | n |
| III-5 | gi│312169310 | Adenosylhomocysteinase (EC 3.3.1.1) | 52.7 | 5.9 | 247.0 | 29 | 69.2 | n |
| III-6 | gi│312166966 | Chaperone protein DnaK | 69.7 | 4.9 | 222.0 | 30 | 48.2 | n |
| III-7 | gi│312168888 | 60 kDa chaperonin GroEL | 57.4 | 5.1 | 97.7 | 16 | 41.0 | n |
| III-8 | gi│312169323 | S-adenosylmethionine synthetase (EC 2.5.1.6) | 42.8 | 4.8 | 182.0 | 20 | 63.8 | n |
| III-9 | gi│312167529 | Protein translation elongation factor Tu (EF-Tu) | 43.1 | 5.2 | 148.0 | 22 | 66.9 | n |
| III-10 | gi│312167773 | Chitosanase (EC 3.2.1.132) | 39.6 | 6.5 | 156.0 | 17 | 63.6 | y |
| III-11 | gi│312167773 | Chitosanase (EC 3.2.1.132) | 39.6 | 6.5 | 127.0 | 16 | 54.0 | y |
| III-12 | gi│312168091 | Chitinase (EC 3.2.1.14) | 42.1 | 8.7 | 42.8 | 8 | 29.1 | y |
| III-13 | gi│312168185 | Peptidyl-prolyl cis-trans isomerase (EC 5.2.1.8) | 28.7 | 8.9 | 91.0 | 12 | 40.8 | y |
| III-14 | gi│312168534 | Chitin-binding protein | 27.0 | 7.7 | 101.0 | 12 | 59.8 | y |
| III-15 | gi│312168534 | Chitin-binding protein | 27.0 | 7.7 | 60.6 | 5 | 34.8 | y |
| III-16 | gi│312167051 | Superoxide dismutase (EC 1.15.1.1) | 23.5 | 5.9 | 128.0 | 9 | 58.7 | n |
| III-17 | gi│312168185 | Peptidyl-prolyl cis-trans isomerase (EC 5.2.1.8) | 28.7 | 8.9 | 84.4 | 13 | 37.6 | y |
| III-18 | gi│312167149 | Inorganic pyrophosphatase (EC 3.6.1.1) | 19.4 | 4.8 | 117.0 | 12 | 84.1 | n |
| III-19 | gi│312167795 | 34 kDa membrane antigen precursor | 21.8 | 6.8 | 85.5 | 13 | 60.7 | y |
| IV-1 | gi│312169059 | Glutamate/aspartate-binding protein | 32.4 | 9.6 | 151.0 | 23 | 62.6 | y |
| IV-2 | gi│312169059 | Glutamate/aspartate-binding protein | 32.4 | 9.6 | 98.6 | 15 | 45.5 | y |
| IV-3 | gi│312167773 | Chitosanase (EC 3.2.1.132) | 39.6 | 6.5 | 71.3 | 8 | 23.0 | y |
| IV-4 | gi│312168534 | Chitin-binding protein | 27.0 | 7.7 | 95.3 | 7 | 37.3 | y |

as a means of entering the cells. To monitor the bacterial invasion of the fungus, we performed several microscopic investigations using GFP-labeled bacteria. As early as 1 day after the infection, when spore formation is not yet visible, confocal laser scanning microscopy revealed that the bacteria were inside the fungal hyphae (*Figure 1*). We then used cryo-electron microscopy to capture the symbionts in the act of infection. This technique allows for a relatively low disturbance of the sample and few artifacts.

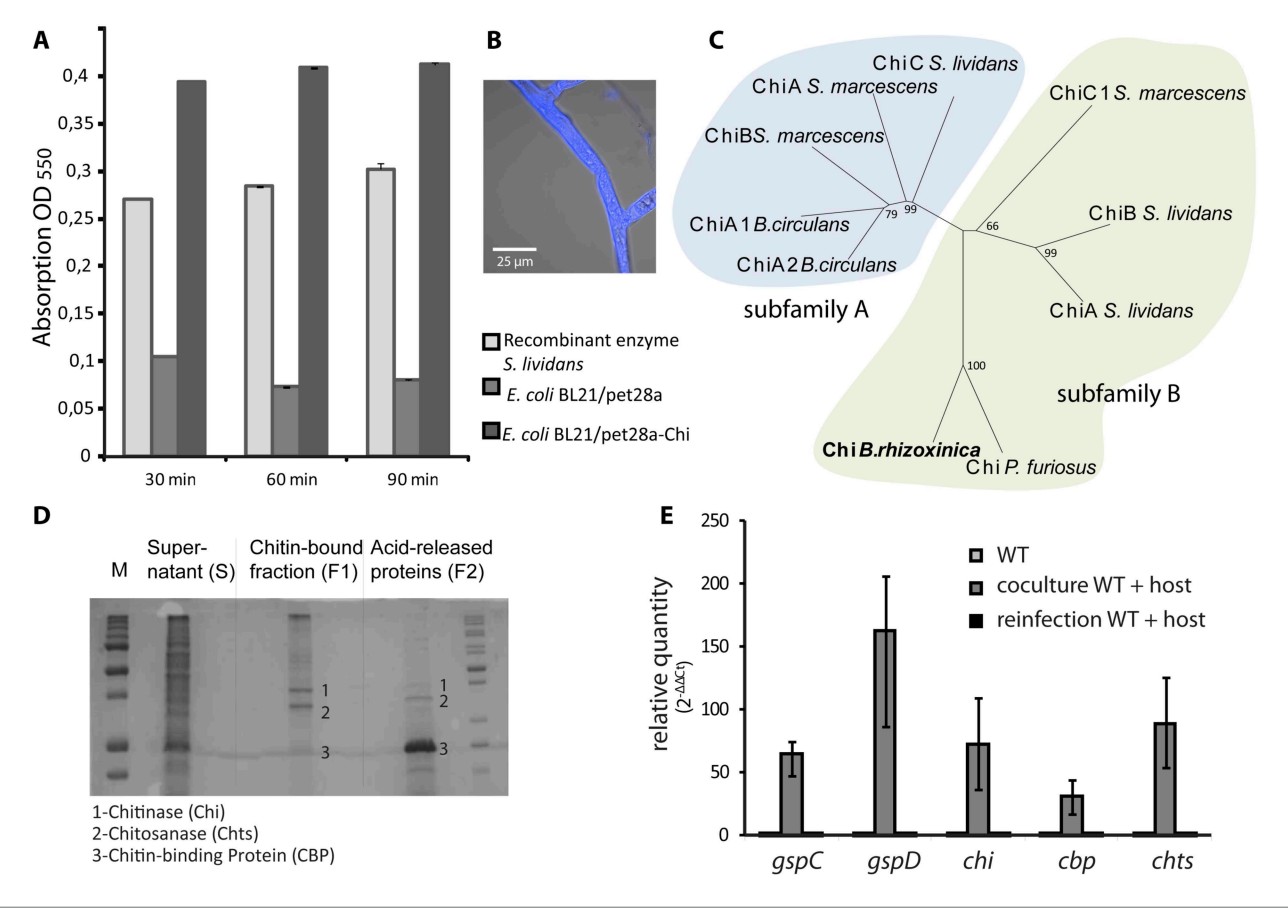

**Figure 5**. Functional analyses of chitinolytic enzymes. (**A**) Chitinase activity in cell-free culture supernatants of an *E. coli* harboring recombinant chitinase from *B. rhizoxinica* after incubation for 30, 60, and 90 min, in comparison to the activity of a recombinant *S. lividans* enzyme. Error bars indicate the standard deviation of three individual experiments. (**B**) Calcofluor staining of *R. microsporus* hyphae. (**C**) Phylogenetic analysis of chitinases from the GH family 18; protein sequences were retrieved from NCBI and comprise subfamily A and B sequences. (**D**) Chitin-binding assay performed using acid released crab shell chitin and the supernatant of *B. rhizoxinica* wt. SDS page of the non-bound fraction (S), the bound protein fraction (F1) and the pelleted chitin with the rest of the bound protein (F2). The three indicated proteins were identified using MALDI-TOF. (**E**) Gene expression assay for T2SS and chitinolytic proteins. The expression of T2SS genes *gspC* and *gspD* as well as *chi, cbp* and *chts* in *B. rhizoxinica* were monitored using RT qPCR in pure culture, in co-cultivation with a cured host (*R. microsporus*) and after re-infection of the cured host. The gene *rpoB* was used as an internal standard for the calculation of expression levels and normalization. The expression of all five genes is substantially increased in the wt during co-cultivation, while expression levels after re-infection decreased to nearly the level in pure wt culture. Error bars indicate standard deviation.

The following figure supplement is available for figure 5:

**Figure supplement 1**. Multiple alignment of the chitinase protein sequences was performed based on the three-dimensional structure of the ChiB sequence from *S. marcescens* (pdb1e15) and Chi from *P. furiosus* (2dsk).

The micrographs permitted an image of clearly distinguishable fungal hyphae (Rm) and a large number of bacteria (Br) surrounding or attaching to them. Fungal hyphae can be seen with a very smooth surface where single bacteria or bacterial colonies are attaching to it (*Figure 6A–E*). Yet at this point, the attachment seems to be purely superficial, and both organisms can still be clearly distinguished from one another. A tight attachment occurs as soon as 1 hr after the co-incubation of the bacteria and fungus. We observed fibrillar structures connecting the bacteria to the fungal surface (*Figure 6D*). In addition to this, we noted pleomorphism of the bacteria (irregular shapes in *Figure 6C* inset and in *Figure 6D,E*). At a later stage (*Figure 6E–H*), the bacteria seem to lose their sharp, pronounced form and enter the fungal cells by fusing with their cell wall (*Figure 6E–H*). After 20 hr of co-culture fungal hyphae appear to lose some of their form and structural integrity. The fusion sites are still clearly visible even though some of the structures can only be vaguely identified as bacteria

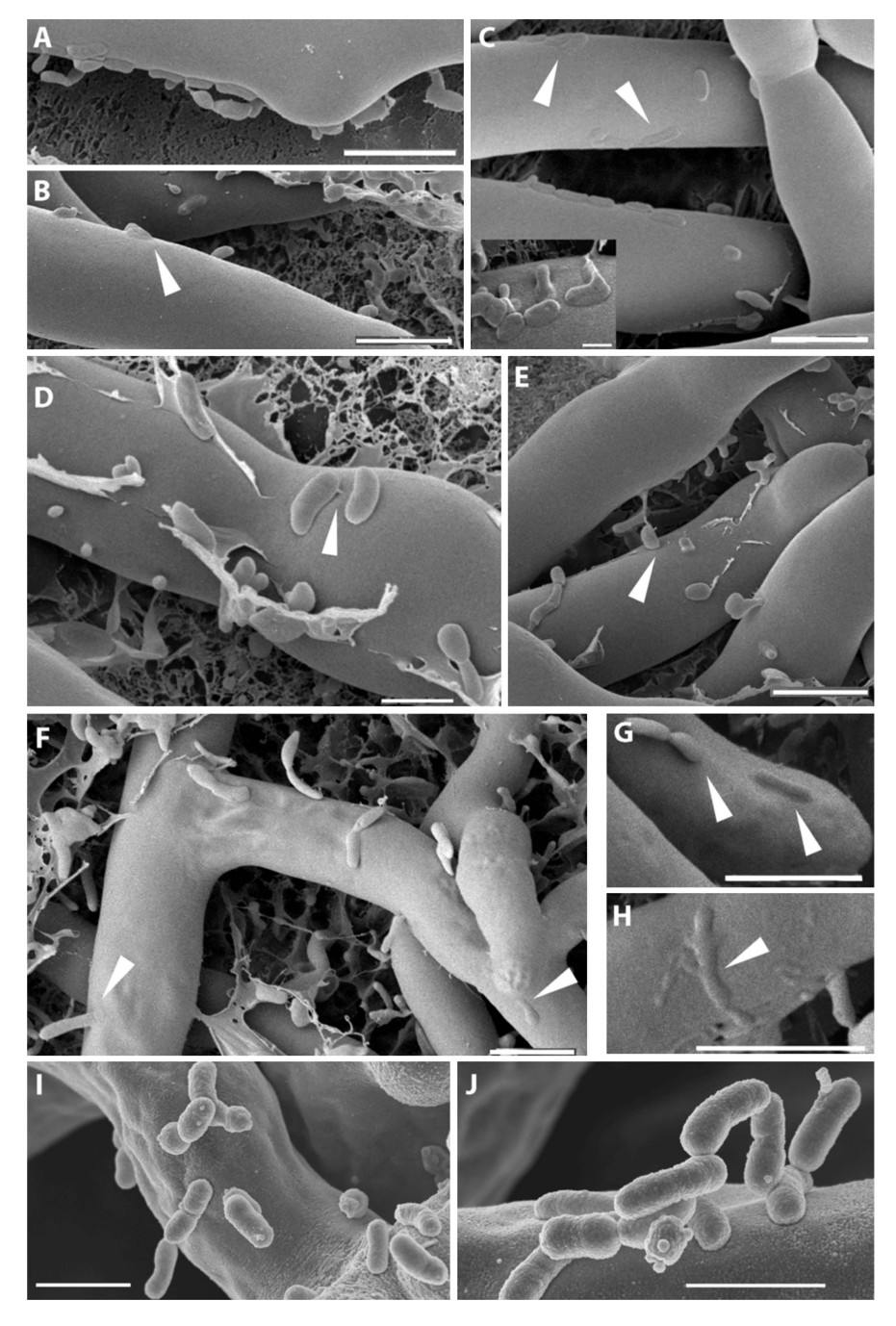

**Figure 6**. Course of infection of *B. rhizoxinica* (Br) to *R. microsporus* (Rm) observed by scanning cryo-electron microscopy. (**A**–**D**) Attachment/adherence of bacteria to fungal hyphae after 1 hr and (**F**–**H**) 20 hr of co-cultivation; (**C**–**D**, **F**–**H**) bacterial and fungal cell walls start to merge; (**D**) fibrillar structures connecting a bacterial cell to the hyphal surface; (**E**–**H**) fusion of cell walls and the intrusion of bacterial cells into the fungal hyphae. White arrows mark areas of particular interest. (**I**–**J**) Scanning electron microscopy of a co-culture of sterile *R. microsporus* with *B. rhizoxinica* Δ*gspD* (**I**) and *B. rhizoxinica* Δ*chi* (**J**). Attachment of both mutant strains to the hyphal surface is visible, however, no intimate contact or fusion events could be observed. Scale bars represent 5 μm.

(*Figure 6C* small image and *Figure 6H*). In some cases, bacteria were caught sticking halfway through the cell wall (*Figure 6E,F*). By arranging the single steps in sequence we obtained snapshots of the complete course of the infection.

Notably, no active engulfment of the bacterium by the fungus has been observed by cryo EM. To further rule out a scenario involving endocytosis we employed fluorescent staining that would permit visualizing endocytosis and vesicular traffic of the fungus. We stained the fungal membrane with styryl dye FM4-64 (Invitrogen, Carlsbad, USA) and the bacterial symbionts by bacteria-specific dye Syto 9 (Invitrogen, Carlsbad, USA). After 1 hr of co-incubation, we observed bacteria attached to the fungus but no fungal membrane surrounding the bacterium. After 5 hr, we detected bacteria within the hyphae, and again no fungal membrane was visible (*Figure 7*).

To evaluate the effect of the genes crucial for hyphal entry, we conducted scanning electron microscopy studies of *B. rhizoxinica* Δ*gspD*::Kanʳ and Δ*chi*::Kanʳ. We screened several co-cultures and detected bacterial cells attached to the surface of fungal hyphae. Nevertheless, it should be pointed out that the bacteria appeared to be only loosely attached, and even after extended co-incubation no fusion events could be detected (*Figure 6I,J*).

## Discussion

*B. rhizoxinica* and *R. microsporus* form a phytopathogenic alliance that jointly produces and secretes the highly potent phytotoxin rhizoxin, the virulence factor inducing rice seedling blight (*Scherlach et al., 2012*). Symbiosis factors such as the *hrp* locus of *B. rhizoxinica* and the LPS layer contribute to the persistence of the tight association of the host fungus and its specific bacterial endosymbiont (*Leone et al., 2010*; *Lackner et al., 2011a*). Although *B. rhizoxinica* has undergone significant genome reduction (*Moran et al., 2008*; *Lackner et al., 2011b*) it still retains the ability to grow in pure culture and to re-infect the sterile fungal host. During this process the bacteria have to penetrate the fungal cell wall barrier. Transmission electron microscopy and freeze-fracture electron microscopy have showed that the endobacteria are not surrounded by a fungal membrane (*Partida-Martinez and Hertweck, 2005*; *Partida-Martinez et al., 2007a*, *2007c*), which rules out an phagocytosis-like vesicular uptake, as seen in the *Nostoc punctiforme—Geosiphon pyriformis* symbiosis (*Mollenhauer et al., 1996*). In this paper we have unveiled an alternative avenue for an active bacterial invasion of fungal hyphae involving the secretion of chitinolytic enzymes.

Based on genomic and proteomic analyses we have discovered a type 2 secretion system in the fungal endosymbiont *B. rhizoxinica* that is central to the *Burkholderia-Rhizopus* interaction. Core components of the T2SS were targeted for deletion and corresponding mutants were incapable of forming a symbiosis. Previous mutational studies of various T2SS have provided evidence of their involvement in pathogenesis (*DeShazer et al., 1999*; *Ali et al., 2000*; *Roy Chowdhury and Heinemann, 2006*). T2SS may also be absent in pathogens, and several T2SS of mutualists have already been described

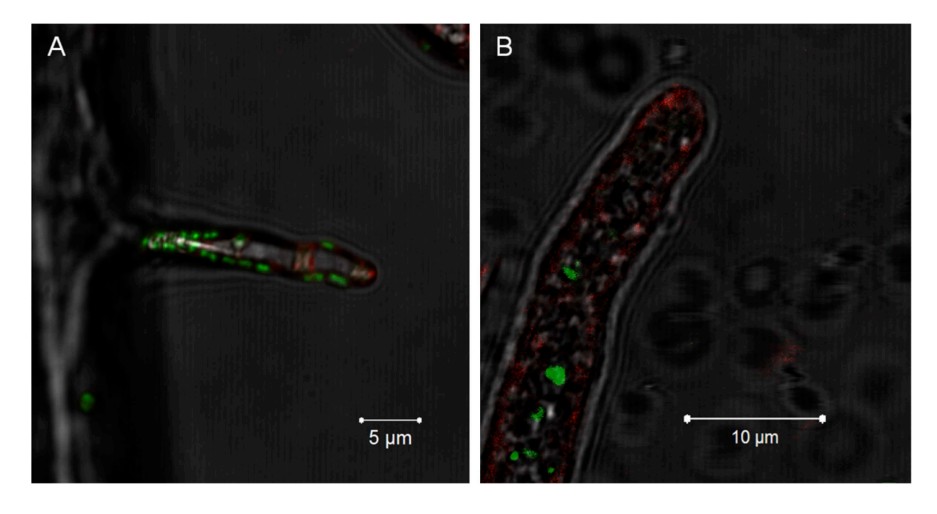

**Figure 7**. Evidence for the lack of active engulfment of *B. rhizoxinica* by *R. microsporus*. (**A**) Bacteria (green) attaches to fungal hyphae within 1 hr. (**B**) The endocytotic activity of the fungus can be observed by the red vesicles that are present all around the hyphae and highly accumulated at the apical tip. After the infection, no fungal membrane is visible around the bacterium.

(*Filloux, 2004*; *Cianciotto, 2005*). However, they are absent in some well-studied aphid and insect symbionts (*Cianciotto, 2005*). In a recent report on the T2SS of the obligate arbuscular mycorrhizal fungus symbiont *Candidatus* Glomeribacter gigasporarum, it was shown that the expression of the gene coding for GspD was up-regulated in the obligate symbiont (*Ghignone et al., 2012*). Here, we demonstrate for the first time that a T2SS is crucial for a bacterial-fungal symbiosis. We also elucidate the key role of the T2SS in secreting chitinolytic enzymes and chitin-binding proteins.

Chitin is well known as one of the major structural components of the fungal cell wall (*Gooday, 1990*; *Adams, 2004*), and chitinases are secreted by bacteria primarily during mycophagy (*de Boer et al., 2004*; *Leveau and Preston, 2008*) and pathogenesis (*Chernin et al., 1995*; *Connell et al., 1998*; *Francetic et al., 2000*). In this study we found that the deletion of the chitinase (*chi*) gene completely abolished the bacteria's ability to enter the fungal hyphae and thus rendering it incapable of establishing a functional symbiosis. The deletion of two additional genes coding for a chitin-binding protein and a chitosanase showed no effect on the sporulation assay. However, these two enzymes are present in great abundance in the *B. rhizoxinica* wt secretome and likely support the action of chitinase. This idea of these enzymes' function is further supported by the fact that the chitosan is highly abundant in the cell wall of zygomycetes (and fungi in general) and that secretion of both proteins is highly reduced in the T2SS mutants, which are unable to intrude the fungus.

Expression levels of these proteins are highly increased in co-culture with the fungus similar to the chitinase gene, suggesting a co-regulation of the transcription of the three genes. Chitin binding protein (Cbp) is the most abundant protein in the cell-free supernatant of *B. rhizoxinica*, and its expression levels are higher in co-culture with the host fungus. We therefore assume that Cbp facilitates bacterial attachment to the fungal hyphae and renders the chitin matrix more accessible to chitinase degradation as it was proposed for family 33 Cbps. Small Cbps also promote the recognition and degradation of chitin by streptomycetes (*Schrempf, 2001*). Chitin-binding proteins could play a role in various close interactions where bacteria attach to the hyphal surface and form fibrillar structures, as in specific *Streptomyces-Aspergillus* (*Siemieniewicz and Schrempf, 2007*) and *Paenibacillus-Fusarium* co-cultures (*Dijksterhuis et al., 1999*). Chitin binding may also set the stage for intrusion, as observed in *Burkholderia* spp. and AM fungal spores (*Levy et al., 2003*) and *Rhizopus* hyphae, as reported in this study.

In light of the fact that chitosan is the dominant component of the Zygomycete cell wall (*Bartnicki-Garcia and Nickerson, 1962*) it is surprising that only the chitinolytic enzyme plays a crucial role in the active invasion of bacteria into fungal cells. However, chitosanase likely supports the invasion process. To establish an intimate association, the physical contact must occur at the right time and the right place and may be dependent upon many factors (*Bright and Bulgheresi, 2010*). The microscopic images and the gene expression studies indicate that the bacteria attach themselves to the fungus even before the chitinolytic enzymes are produced and secreted. In this way low concentrations of lytic enzymes would be sufficient for local activity. Thus, fungal cell wall penetration is a more melting-like, mild process without damaging the hyphae. A similar scenario has been described in the context of plant infection, where the precise and highly localized cellulolytic activity of cellulase CelC2 from *Rhizobium leguminosarum* bv. *trifolii* degrades the host plant cell wall during penetration (*Robledo et al., 2008*).

The model of topical cell wall lysis in the *Burkholderia-Rhizopus* interaction is supported by the microscopic snapshots of the progress of hyphal colonization and intrusion. As early as 1 hr after co-incubation with fungal hyphae, a close attachment of the bacterial cells can be observed, followed by fusion with the fungal cell wall. This process is observed very locally for every bacterium, even when several bacteria form fusion structures close to each other on the fungal cell wall. Although parts of the penetrated cell wall may appear a bit irregular (*Figure 6F*), there are no visible signs of cell lysis or loss of integrity surrounding the intrusion sites. Both, the mutants lacking chitinase or T2SS components attach in a comparable yet weaker fashion to the fungal hyphae but are not capable of similar fusion events (*Figure 6I–J*). Moreover, cell membrane staining to fungus has shown that the fungus does not engulf the bacterium by endocytosis. Overall, this strategy permits a traceless entry into the fungal cells, thus guaranteeing that host integrity is not affected.

In summary, we identified a T2SS and a secreted chitinase as two molecular mechanisms involved in the attachment and infection process of an agricultural and medicinal relevant bacterial-fungal interaction. Secretion of chitinase and presumably further effector proteins translocated via a T2SS help to locally soften the fungal cell wall allowing bacterial entry and preventing the disintegration of fungal hyphae (*Figure 8*). Considering the growing number of reports about endobacteria in mycorrhiza and other fungi (*Bianciotto et al., 2000*; *Bonfante and Anca, 2009*; *Kobayashi and Crouch, 2009*;

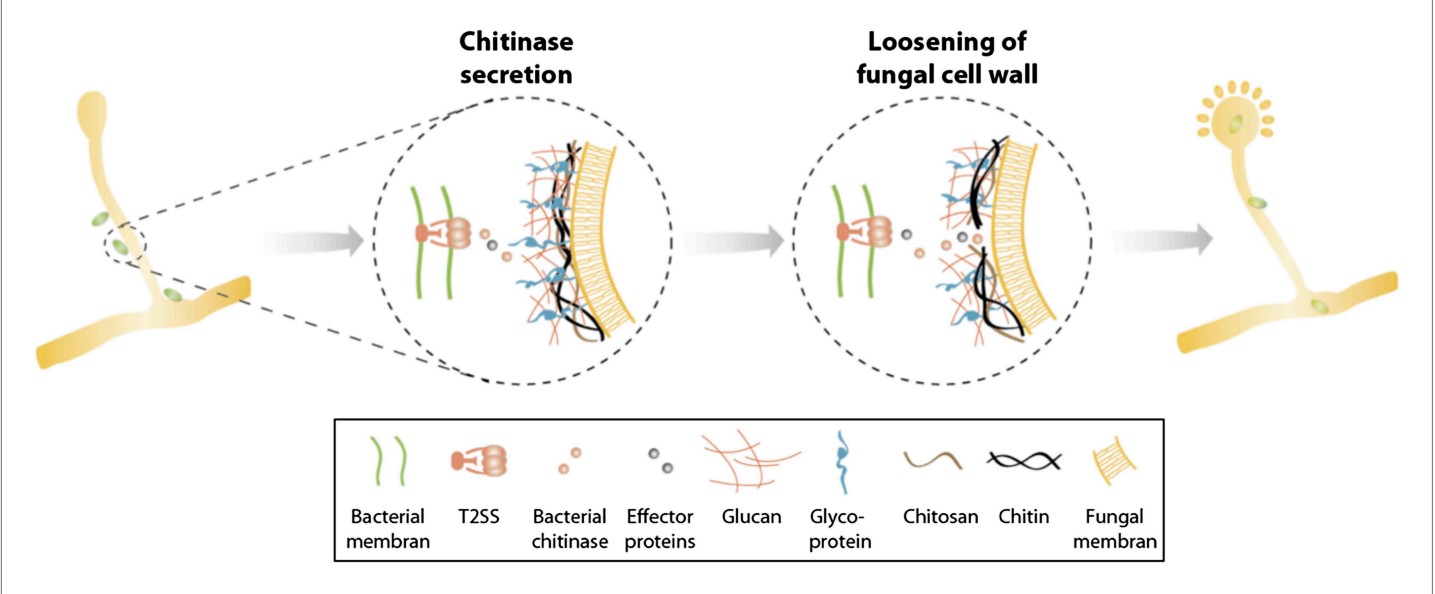

**Figure 8**. Model of processes involved in bacterial invasion. Chitinase as well as other effector proteins are secreted via a bacterial T2SS and induce a local dissolution of the fungal cell wall. This enables bacteria to enter and colonize the fungal cell and induce sporulation.

*Hoffman and Arnold, 2010*; *Frey-Klett et al., 2011*) as well as the first indications about their functional implication in the bacterial-fungal and plant-fungal relationships (*Partida-Martinez and Hertweck, 2005*; *Sharma et al., 2008*) it is striking that there is such a lack of knowledge about the acquisition and the establishment of such associations.

As the fates of bacteria and fungi are ecologically intimately connected in soil one can envision that the endosymbiotic associations could be much more widespread in nature. Indeed, the genetic repertoire for chitinolytic enzymes and a T2SS can be found in other endobacteria, implying an involvement of these systems in other bacterial-fungal interactions. Bacteria from the order *Burkholderiales* are among the most frequently identified intracellular bacteria in fungal hyphae (*Hoffman and Arnold, 2010*; *Frey-Klett et al., 2011*). Thus, our findings could present a model system for many other horizontally acquired symbionts and might help deepen the understanding of the common mechanisms involved in the interaction of proteobacteria with eukaryotic cells. Overall, this is the first report of the molecular basis of bacterial invasion of a fungus and the first visualization of the invasion process. We believe that the mechanisms employed are widespread and occur in the growing number of known bacterial-fungal endosymbioses as an alternative pathway to endocytosis (vesicular uptake).

## Material and methods

### Strains and culture conditions

*Burkholderia rhizoxinica* (isolate B1, HKI 0454) and the *Rhizopus microsporus* (ATCC62417) harboring endobacteria as well as the symbiont free *R. microsporus* (ATCC62417/S) (*Partida-Martinez et al., 2007c*) were used in this study. Pure cultures of *B. rhizoxinica* were grown in MGY medium (M9 minimal medium supplemented with 1.25 g l⁻¹ yeast extract and 10 g l⁻¹ gycerol) at 30°C or TSB M9 medium (10 g l⁻¹ glycerol, 3 g l⁻¹ yeast extract, 15 g l⁻¹ tryptone soy broth, M9 salts) respectively. Wild type and cured strains of *R. microsporus* were grown on Potato Dextrose Agar (PDA) at 30°C or in TSB respectively. All the strains used in this study are listed in *Table 3*.

### Generation of T2S deficient and chitinolytic-deficient mutants of *B. rhizoxinica*

To address the involvement of both T2SS and chitinolytic proteins two genes of the type 2 secretion gene cluster as well as the gene annotated as chitinase were targeted by double crossover using a suicide-vector harboring a mutated phenylalanyl tRNA synthetase gene *pheS* as a counter-selectable

**Table 3.** Bacterial and fungal strains

| Stains | Characteristics | References |
|---|---|---|
| *Burkholderia rhizoxinica* HKI-0454 | Wild type, isolated from *Rhizopus microsporus* ATCC62417 | * |
| *Rhizopus microsporus* ATCC62417 | Fungal host harboring bacterial endosymbionts, isolated from rice seedlings | † |
| *B. rhizoxinica* Δ*gspD*::Kanʳ | T2SS mutant *B. rhizoxinica* with deletion of *gspD* | This study |
| *B. rhizoxinica* Δ*gspC*::Kanʳ | T2SS mutant *B. rhizoxinica* with deletion of *gspC* | This study |
| *B. rhizoxinica* Δ*chit*::Kanʳ | *B. rhizoxinica* with deletion of chitinase gene | This study |
| *B. rhizoxinica* Δ*cbp*::Kanʳ | *B. rhizoxinica* with deletion of chitin-binding protein gene | This study |
| *B. rhizoxinica* Δ*chts*::Kanʳ | *B. rhizoxinica* with deletion of chitosanase gene | This study |
| *B. rhizoxinica*/pHKT4 | *B. rhizoxinica* wt harboring a RFP expression vector | This study |
| *B. rhizoxinica* Δ*gspD*::Kanʳ/pHKT2 | *B. rhizoxinica gspD* mutant harboring a GFP expression vector | This study |
| *B. rhizoxinica* Δ*gspC*::Kanʳ/pHKT2 | *B. rhizoxinica gspC* mutant harboring a GFP expression vector | This study |
| *B. rhizoxinica* Δ*chit*::Kanʳ/pHKT2 | *B. rhizoxinica chit* mutant harboring a GFP expression vector | This study |
| *B. rhizoxinica* Δ*cbp*::Kanʳ/pHKT2 | *B. rhizoxinica cbp* mutant harboring a GFP expression vector | This study |
| *B. rhizoxinica* Δ*chts*::Kanʳ/pHKT2 | *B. rhizoxinica chts* mutant harboring a GFP expression vector | This study |
| *R. microsporus* + *B. rhizoxinica* Δ*gspD*::Kanʳ/pHKT2 | Reinfected cured fungal host with *gspD* mutant harboring a GFP expressing vector | This study |
| *R. microsporus* + *B. rhizoxinica* Δ*gspC*::Kanʳ/pHKT2 | Reinfected cured fungal host with *gspC* mutant harboring a GFP expressing vector | This study |
| *R. microsporus* + *B. rhizoxinica* Δ*chit*::Kanʳ/pHKT2 | Reinfected cured fungal host with *chit* mutant harboring a GFP expressing vector | This study |
| *R. microsporus* + *B. rhizoxinica* Δ*cbp*::Kanʳ/pHKT2 | Reinfected cured fungal host with *cbp* mutant harboring a GFP expressing vector | This study |
| *R. microsporus* + *B. rhizoxinica* Δ*chts*::Kanʳ/pHKT2 | Reinfected cured fungal host with *chts* mutant harboring a GFP expressing vector | This study |
| *Escherichia coli* BL21(DE3)/ pET28a-Chi | *E. coli* with expression vector harboring chitinase gene | This study |

*__Partida-Martinez and Hertweck (2005)__ Pathogenic fungus harbours endosymbiotic bacteria for toxin production. Nature 437:884–888.

†__Ibaragi (1973)__ Studies on rice seedling blight. I. Growth injury caused by *Rhizopus* sp. under high temperature. Ann. Phytopathol. Soc. Jpn 39:141–144.

marker as described previously with slight modifications (**Lackner et al., 2011a**). Flanking regions upstream and downstream of the selected gene were amplified using a proof-reading polymerase with primers containing 20 bp homologous to the flanking region and 20 additional bp targeting the 3' and 5' end of a kanamycin cassette. The same 20 bp have been used for primers amplifying a kanamycin cassette from pK19 as template. A triple overlapping PCR was performed with equimolar amounts of the flanking PCR fragments and a twofold amount of the kanamycin cassette PCR product using PhusionFlash High-Fidelity PCR Master Mix (Thermo Fisher Scientific, Waltham, USA). The yielded fragments were subsequently transformed into pCR4Blunt-TOPO vector (Invitrogen, Paisley, UK) and after restriction digest ligated into pGL42a. Knockout constructs for cbp and chts were obtained following the previous described amplification method (**Lackner et al., 2011a**). The vectors pNM89 (GspC), pNM91 (GspD), pZU02 (Chi), pGL47 (Chts) and pGL49 (Cbp) were introduced into competent cells of *B. rhizoxinica* by electroporation. Transformants were selected on standard nutrient agar supplemented with 50 µg ml⁻¹ kanamycin. Colonies were inoculated in liquid MCGAVT medium (**Lackner et al., 2011a**) for 3 to 4 days and subsequently spread on MCGAVT agar plates to obtain single colonies. This procedure was repeated several times and obtained clones were checked for correct integration of the knockout construct into the genome via PCR targeting an internal fragment of the respective gene (int) the *pheS* gene (pheS) as well as a region spanning the two recombination sites with ArmA and ArmB amplifying wt fragments and ArmC and Arm D mutant fragments, respectively. The complete list of primers used in this study is found in *Table 4*.

**Table 4.** Primers used in this study

| | Name | Oligo sequence |
|---|---|---|
| Primers used for generating KO mutants | TII_D_fl1_fw | 5′-GCTACGGATCCCTGCCAGGTATTGCCGTATT-3′ |
| | TII_D_fl1_rv | 5′-GCTACAAGCTTCAATCAGCTTGTCGAATTGC-3′ |
| | TII_D_fl2_rv | 5′-GCCCAGTAGCTGACATTCATCCCCGATCAATTATGCAAGCAG-3′ |
| | TII_D_fl2_fw | 5′-TTCTTGACGAGTTCTTCTGATGGACTGGATGTCTGGATCA-3′ |
| | TII_C_fl1_fw | 5′-GCTACGAATCTCAGATCTGTGCGAGGATTG-3′ |
| | TII_C_fl1_rv | 5′-GCTACAAGCTTCAACTCGCCTTTACGTACCC-3′ |
| | TII_C_fl2_rv2 | 5′-GCCCAGTAGCTGACATTCATCCGACGGCATGATGAGTTTGTG-3′ |
| | TII_C_fl2_fw2 | 5′-TTCTTGACGAGTTCTTCTGAAGCAAGCTGGTCAGGAACAT-3′ |
| | Kan_F | 5′-ATGATTGAACAAGATGGATTGC-3′ |
| | Kan_R | 5′-GCCTTCTTGACGAGTTCTTCTGA-3′ |
| | Chi_Fl1_F | 5′-GAACTAGTCTCGATCATGGGGGTATTTG-3′ |
| | Chi_Fl1_R | 5′-GCCCAGTAGCTGACATTCATCCCAGGTGCTTTTTCATTGCTTC-3′ |
| | Chi_Fl2_F | 5′-GCCTTCTTGACGAGTTCTTCTGACGTGACGTATCGTGCAAAGT-3′ |
| | Chi_Fl2_R | 5′-ATCCCGGGACGCGGTCAAGTCGATGTAG-3′ |
| | Kan_Chi_F | 5′-GAAGCAATGAAAAAGCACCTGGGATGAATGTCAGCTACTGGGC-3′ |
| | Kan_Chi_R | 5′-ACTTTGCACGATACGTCACGTCAGAAGAACTCGTCAAGAAGGC-3′ |
| | P1_chtos | 5′-GCTACGGGCCCGGCATCGGTGACTATCGTAAC-3′ |
| | P2_chtos | 5′-GCTACTTAATTAAGCTAGCGTAGCACAGCCGATACCGTAAGC-3′ |
| | P3_chtos | 5′-GCTACGCTAGCTTAATTAAGTAGCGCAATGGAGCAAGCTGATGG -3′ |
| | P4_chtos | 5′-GCTACGCGGCCGCAACGTGCGCGACGATACGTTC-3′ |
| | Kan_chts_F | 5′-GGATGAATGTCAGCTACTGGGC-3′ |
| | Kan_chts_R | 5′-TCAGAAGAACTCGTCAAGAAGGC-3′ |
| | P1_chtbdp | 5′-GCTACGCGGCCACGCCGAGATGATGTTG-3′ |
| | P2_chtbdp | 5′-GCTACTTAATTAAGCTAGCGTAGCCGATCGTGCGTGAGTAAG-3′ |
| | P3_chtbdp | 5′-GCTACGCTAGCTTAATTAAGTAGCAGCCAACCGACGTACCTACC-3′ |
| | P4_chtbdp | 5′-GCTACGGGCCCAAGACGGCGGGCGTATTACC-3′ |
| | Kan_cbp_F | 5′-GGATGAATGTCAGCTACTGGGC-3′ |
| | Kan_cbp_R | 5′-TCAGAAGAACTCGTCAAGAAGGC-3′ |
| Primers used for RT-qPCR studies | TIISS_D_RT_F | 5′-GAGCAGCGATACCAACATCC-3′ |
| | TIISS_D_RT_R | 5′-TTGAATGCGGAGACCGAAG-3′ |
| | TIISS_C_RT_F | 5′-AGCGTCACTTACTGGGTCATC-3′ |
| | TIISS_C_RT_R | 5′-CGAGCCGAACAGAGTTTGAG-3′ |
| | Chi_RT_F | 5′-CGCTGGATACGGTCAACATC-3′ |
| | Chi_RT_R | 5′-GCCTTGCACGTCATTCTT-3′ |
| | CBP_RT_F | 5′-ACGACAGCGCATAATCCTTC-3′ |
| | CBP_RT_R | 5′-GGGTGCATCGTAAATCAGGT |
| | Chtos_RT_F | 5′-AGGTGGACTGACCCGTATTG |
| | Chtos_RT_R | 5′-TTGCACGCTGTATTGGATGT-3′ |
| | rpoB_RT_F | 5′-ATTTCCTTCACCAGCACGTT-3′ |
| | rpoB_RT_R | 5′-TTCGGGGAAATGGAAGTGT-3′ |

*Table 4. Continued on next page*

Table 4. Continued

| | Name | Oligo sequence |
|---|---|---|
| Primers used for control of generated mutants | Arm_A_rv (Kan) | 5'-AGTGACAACGTCGAGCACAG-3' |
| | Arm_B_fw (Kan) | 5'-CGTTGGCTACCCGTGATATT-3' |
| | TIISSD_C_A_fw | 5'-TCACCTCACGTAGCAGATCG-3' |
| | TIISSD_C_rv | 5'-GCATCGACGAAATTCAAGGT-3' |
| | TIISSD_D_fw | 5'-GATAACCGGATCGTCAAGGA-3' |
| | TIISSD_D_B_rv | 5'-CCGGACAAGTCGTACTCGAT-3' |
| | TIISSD_Int_fw | 5'-GTCGAGGGACCAAAGTTTCA-3' |
| | TIISSD_Int_rv | 5'-GGCGTAGACAGGATGTTGGT-3' |
| | TIISSC_CA_fw | 5'-ACTCCAGCCCGCATACATAC-3' |
| | TIISSC_C_rv | 5'-ATTCAGCGCACGTAGATCGT-3' |
| | TIISSC_D_fw | 5'-GCGTCACTTACTGGGTCATC-3' |
| | TIISSC_DB_rv | 5'-AGGAAGTGCTGCGTGTAACC-3' |
| | Chi_Ctrl1_KO_F | 5'-GAACCATTCGCCTTCTTCAC-3' |
| | Chi_Ctrl1_WT_R | 5'-ATCGCTTTCAACAGGTGCTT-3' |
| | Chi_Ctrl2_WT_R | 5'-CCAGTTGTGGCAAATGATTG-3' |
| | Chi_Ctrl2_KO_R | 5'-ATTTCGGCTCTGACGTGACT-3' |
| | Chi_Int_fw | 5'-TGACCTCCATCGCCAAGTCG-3' |
| | Chi_Int_rv | 5'-CGGAACACCTGCGTGAATGC-3' |
| | Chtos_ext_for_1 | 5'-GAAGCGTGATGTGATTGAAG-3' |
| | Chtos_ext_rev_1 | 5'-AAGTCGCATCCAGACATTG-3' |
| | Chtos_int_for_1 | 5'-GACGCCAAGACGATCTACCA-3' |
| | Chtos_int_rev_1 | 5'-TTGGGCTTTGACCTTGCTAC-3' |
| | Cbp_ext_for_1 | 5'-ACTTTCTGAATACAGCTTGC-3' |
| | Cbp_ext_rev_1 | 5'-CAGTCATGATGCAATACGTG-3' |
| | Cbp_int_for_1 | 5'-GCGGTCTAGTCCCTGCTTAC-3' |
| | Cbp_int_rev_1 | 5'-GAGGCTATTGGTCGTCACCT-3' |
| | pBS_nspl_for_I | 5'-AGCTCACTCAAAGGCGGTAA-3' |
| | pBS_nspl_rev_I | 5'-TTTTTGTGATGCTCGTCAGG-3' |

## Low-temperature scanning electron microscopy (LT SEM)

Low-temperature scanning electron microscopy of uncoated samples was performed as described previously (*Schubert et al., 2007*; *Jennessen et al., 2008*). Fungal mycelium from the cured *R. microsporus* ATCC62417/S was inoculated on NA agar plate and incubated overnight to start hyphal growth. On the edges of hyphal spreading 2 µl of *B. rhizoxinica* culture was spotted and the plate was again incubated for approximately 2 hr. From the interaction zone several parts were selected and excised with a surgical blade as small agar blocks, and transferred to a copper cup for snap-freezing in nitrogen slush. Agar blocks were glued to the copper (KP-Cryoblock, Klinipath, Duiven, Netherlands). Samples were examined in a JEOL 5600LV scanning electron microscope (JEOL, Tokyo, Japan) equipped with an Oxford CT1500 Cryostation for cryo-electron microscopy (cryoSEM). Electron micrographs were acquired from uncoated frozen samples, or after sputter-coating by means of a gold/palladium target for three times during 30 s. Micrographs of uncoated samples were taken at an acceleration voltage of 3 kV, and consisted out of 30 averaged fast scans, and at 5 kV in case of the coated sample.

## Gene expression study

RNA was isolated using the RiboPure Bacteria or RiboPure Yeast Kit (Ambion, Texas) following the manufacturers' instruction either from pure bacterial culture or from cocultivation of *B. rhizoxinica* with sterile fungus. 40 ng of total RNA served as template for one-step RT qPCR using gene specific

primers and Quanta sybr green Kit (Quanta BioSciences, Gaithersburg, MD). Realtime PCR was performed on an Eppendorf realplex mastercyler (Eppendorf, Hamburg, Germany) in triplicate for each sample, and a control reaction without enzyme was included for each sample. The *rpoB* gene was used as an internal standard for calculation of expression levels and normalization. For cycling parameters we followed the manufacturers' protocol. Controls without template were included for each primer pair. Cycle threshold (Ct) values were calculated by the realplex software and used for quantification of expression levels via the $2^{-\Delta\Delta Ct}$ method (*Livak and Schmittgen, 2001*).

## Homology search and structure prediction

Homology search was carried out using the NCBI BlastP and the Burkholderia genome database. Domain prediction using Robetta (*Kim et al., 2004*) and Coils (*Lupas et al., 1991*), visualization and modification was done in SwissPdbViewer and VMD. PROMALS3D was used for structure based alignment of chitinases (*Pei and Grishin, 2007*). The structural model of *B. rhizoxinica* chitinase was generated by threading the amino acid sequence to PDB database and building the model using Pymol.

## Phylogenetic analysis

For phylogenetic analysis, protein sequences were aligned by the ClustalW algorithm implemented in the MEGA 3.1 software package (*Kumar et al., 2004*). The obtained alignment blocks were used for tree-construction by the neighbor-joining method. 10,000 bootstrap replicates were run to estimate reliability of the inferred groups.

## Reinfection/sporulation assay for bacterial-fungal interaction

Bacterial cells were grown in MGY medium without antibiotics. Fungus was cultivated in 8-well plates in TSB-medium. For reinfection/sporulation assay 6-well plates were used. Each well was filled with 5 ml TSB-medium and inoculated with a pellet of *R. microsporus* mycelium from a 48 plate well. 200 µl of bacterial culture was added and incubated at 30°C. After 4–7 days, sporulation of plates was examined by eye.

## Chitin-binding assay

To a volume of 250 ml supernatant of *B. rhizoxinica* wt 15 mg of acid swollen chitin were added and the mixture was stirred for 60 min at room temperature to allow for chitin binding. Subsequently it was centrifuged at 6000×*g* for 10 min to pellet chitin with bound proteins, the supernatant was stored (S1). The pellet was then washed twice with 0.9% NaCl and resuspended in 0.05 M NaCl to remove bound proteins. Chitin was once again centrifuged to give fraction C (pelleted chitin) and 0.1 M Tris–HCl pH 7 was added to the supernatant (S2). All three fractions were loaded on a SDS polyacrylamide gel and bands were identified MALDI-TOF.

## Secretome sample preparation for 2-D gel electrophoresis

250 ml of bacterial cultures and bacterial/fungal cocultures respectively were centrifuged at 8000×*g* at 4°C for 20 min and the obtained supernatant was sterile vacuum filtered (0.2 µM pore size) and supplemented with 10 g l$^{-1}$ of TCA for wt cultures and 50 g l$^{-1}$ for mutants. The mixture was store at 4°C overnight to allow for protein precipitation. Proteins were pelleted by centrifugation at 12,000×*g* at 4°C for 30 min, the supernatant was removed and the pellet was rinsed twice in ice-cold acetone. The pellet was air-dried for 15 min at room temperature and subsequently resuspended in 300 µl 2D-lysis buffer (7 M urea, 2 M thiourea, 2% [wt/vol] CHAPS(3-[(3-cholamidopropyl)-dimethylammonio]-1-propanesulfonate), 1% [wt/vol] Zwittergent 3-10, 30 mM Tris). To improve protein solubility the samples were sonicated for 5 min in an ultrasonic bat. After centrifugation at 14,000×*g* for 20 min at 4°C, the supernatant was collected. The protein concentration was determined according to the Bradford method.

## 2-D gel electrophoresis analysis

For the separation of proteins in the first dimension 11 cm IPG strips with a nonlinear pH range from both pH 3 to 11 (GE Healthcare Bio-Sciences, Uppsala, Sweden) which had been rehydrated overnight (7 M urea, 2 M thiourea, 2% [wt/vol] CHAPS, 1% [wt/vol] Zwittergent 3_10, 0.002% [wt/vol] bromophenol blue, 0.5% [vol/vol] IPG buffer, 1.2% [vol/vol] De-Streak reagent [GE Healthcare Bio-Sciences, Uppsala, Sweden]) were used as described (*Kniemeyer et al., 2006*). Equal amounts of protein samples from *B. rhizoxinica* wt and mutant's pure culture as well as the respective cocultures were applied

via anodic cup loading to IPG strips. Isoelectric focusing was conducted according to the following protocol: 4 hr at 300 V (gradient), 3 hr at 600 V (gradient), 4 hr at 1000 V (gradient), 5 hr at 8000 V (gradient) and 48,000 V hr at 8000 V (step). Subsequently strips were equilibrated for 10 min in 10 ml of equilibration buffer (6 M urea, 30% [vol/vol] glycerol, 2% [wt/vol] SDS (sodium dodecyl sulfate), 75 mM Tris, 0.002% [wt/vol] bromophenol blue) containing 1% (wt/vol) DTT and subsequently for 10 min in 10 ml of equilibration buffer containing 2.5% (wt/vol) iodoacetamide. Ettan DALT System (GE Healthcare Bio-Sciences, Uppsala, Sweden) was used to separate proteins in the second dimension. SDS polyacrylamide gels (Mini-Protean TGX Precast Gels, AnyKD, BIORAD Hercules, CA) were loaded with the strips and run for 50 min at 200 V. In order to identify the proteins by mass spectrometry (MS), the gels were stained with colloidal Coomassie Brilliant Blue according to *Kniemeyer et al. (2006)* followed by manual excision of the spots. Protein spots were tryptically digested (*Shevchenko et al., 1996*). Extracted peptides were measured and identified on an Ultraflex I and Ultraflextreme MALDI-TOF/TOF device using flexControl 3.3 for data collection and flexAnalysis 3.3 spectra analysis/ peak list generation (Bruker Daltonics, Bremen, Germany). Peptide mass fingerprint (PMF) and peptide fragmentation fingerprint (PFF) spectra were submitted to the MASCOT server (MASCOT 2.3, Matrix Science, London, UK), searching the *B. rhizoxinica* database.

## Heterologous chitinase production

25 ml of *E. coli* BL21 strains carrying the vector pET28a or pET28a/chitinase were cultured in TB (12 g l$^{-1}$ tryptone, 24 g l$^{-1}$ yeast extract, 4 ml l$^{-1}$ glycerol) buffered with 100 mM 2-(*N*-morpholino) ethanesulfonic acid (MES) at pH6. The cells were induced at an $OD_{600}$ of 0.6 with 1 mM IPTG, and grown overnight to $OD_{600}$ of around 14. For SDS polyacrylamide gel 500 μl of cultures were centrifuged for 5 min. Supernatant and pellet are separated. Cell pellets were dissolved in 500 μl TB-MES. 30 ml of the samples (supernatant and pellet, respectively) were suspended in 30 μl protein sample buffer. Samples were boiled for 15 min, and 20 μl run on 12% (wt/vol) SDS polyacrylamide gel. Gels were stained with Coomassie blue.

## Recombinant chitinase assay

Chitinase activity was detected using the aqueous solution of CM-Chitin-RBV as a substrate (Loewe Biochemica GmbH, Sauerlach, Germany) as described by (*Saborowski et al., 1993*). The secretome of BL21 cells bearing pET28a and pET28a/chitinase were used to conduct the assay. Cells were harvested and supernatants were filter-sterilized, 250 μl of the secretome was mixed with 250 μl CM-Chitin substrate and buffered with 250 μl 0.1 M sodium acetate at pH6. Triplicates of samples were incubated at 37°C for 15–90 min. An equal amount of 0.1 U *Streptomyces griseus* recombinant chitinase solution (Sigma-Aldrich, St. Louis, MO) was included as a positive control for chitinase activity. Each reaction was stopped by adding 250 μl of 0.1 N HCl and kept on ice for at least 5 min to ensure complete precipitation of the non-degraded substrate a low pH (<3). After centrifugation at 15.000×*g* for 10 min the absorbance of the supernatants was measured photometrically at 550 nm. Blanks without substrate or enzyme where run in parallel.

## Confocal fluorescence microscopy

To visualize the localization of the *B. rhizoxinica* mutant strains with respect to the fungal hyphae, the RFP encoding pHKT4 plasmid was transformed into the *B. rhizoxinica* wt and GFP encoding pHKT2 plasmid were transformed into *B. rhizoxinica* Δ*chit*::Kan$^r$, Δ*gspC*::Kan$^r$ and Δ*gspD*::Kan$^r$ mutant strains. Subsequently all resulting strains were co-cultured with sterile *R. microsporus*. After 3 to 4 days, a small piece of growing fungus from the co-culture as well as from the sterile fungus was examined. The endocytotic capacity of *R. microsporus* has been visualized by styryl dye FM4-64. A piece of freshly growing *R. microsporus* mycelium has been co-incubated with growing *B. rhizoxinica* culture in 500 μl physiological saline. After 50 min 3 μM of FM4-64 and 5 μM of Syto9 has been added and incubated for 10 min. The live images have been taken in mounted slide to avoid drying. All images have been taken by using a Zeiss CLSM 710 confocal laser-scanning microscope (Göttingen, Germany) for fluorescence detection.

## Acknowledgements

We would like to thank Maria Poetsch and Tom Bretschneider for the MALDI measurements and Dr Olaf Kniemeyer for his advice on proteome analyses. This research was financially supported by the *Jena School for Microbial Communication* (JSMC).

## Additional information

### Funding

| Funder | Author |
|---|---|
| Jena School for Microbial Communication | Nadine Moebius, Zerrin Üzüm |

The funder had no role in study design, data collection and interpretation, or the decision to submit the work for publication.

### Author contributions

NM, ZÜ, JD, Acquisition of data, Analysis and interpretation of data, Drafting or revising the article; GL, Conception and design, Analysis and interpretation of data; CH, Conception and design, Drafting or revising the article

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
