## [Decision Letter]

Thank you for sending your work entitled “Active Invasion of Bacteria into Living Fungal Cells” for consideration at *eLife.* Your article has been favorably evaluated by Richard Losick (Senior editor) and 3 reviewers, one of whom, Thorsten Nuernberger, is a member of our Board of Reviewing Editors.

The Reviewing editor and the other reviewers discussed their comments before we reached this decision, and the Reviewing editor has assembled the following comments to help you prepare a revised submission.

All reviewers acknowledge that your study provides initial mechanistic insight into how bacterial endophytes might enter fungal hosts and might accommodate themselves therein. Due to the rather general principle that you propose it is likely that this entry mechanism holds true for many bacteria-fungus symbioses. Your contribution is thus considered to be of interest to a broad readership. All reviewers however feel that your manuscript requires a revision in which you may address the issues listed below:

1) It is desirable that the authors provide causal evidence that it is indeed chitinase activity that is required for infection. This is of particular importance as the authors themselves stated that supernatants of bacterial cultures contained rather little chitinolytic activity (how much?). The authors may thus consider answering any of the following questions: Is the bacterial chitinase identified indeed capable of loosening the cell wall of this fungus? Would chitinase mutants lacking enzymatic activity still be able to support symbiosis formation? Would chitinase inhibitors be able to block or slow down bacterial entry?

2) There is no mention on the mechanism of entry after cell wall lysis as bacteria still have to pass the fungal membrane. Given that in previous studies it was shown that bacteria are not encapsulated by a host membrane after entering the fungus, they should penetrate the membrane in a different way. Are there examples for that? Perhaps Co-staining of the membrane (FM4-64) and the bacteria (GFP) could help to solve the question. Perhaps there is endocytosis but the membrane dissolves quickly afterwards? The reference cited in the text (46) does, in my view, not rule out the possibility of endo/phagocytosis.

3) The manuscript requires revision in order to make it understandable for a broader readership. In particular, more precision is required regarding the description of the process at a cellular level and correct terms have to be used.

a) A schematic picture describing the (putative) infection process and the final stage of endosymbiosis would be very helpful for the reader. It would make it clear that the EM pictures provided by the authors address the penetration of the cell wall and not the uptake of the bacterium by the fungal cell. Unfortunately, the authors frequently mix up these terms and they should be more precise in that respect.

b) The authors are not precise in the use of “mycelium” and “hyphae”. The bacteria infect the hyphae and not the mycelium. The authors state that “bacteria ... enter the fungal cell by fusing with their cell wall”. It is not clear what fusion with the cell wall means and how such a process could lead to entry of the fungal cell (the cell wall is extracellular!).

c) In Figure 6–figure supplement 1 the authors show a model where chitinase is secreted by the T2SS. If this reviewer understood the manuscript correctly, chitinase is secreted by the classical secretion machinery and not by T2SS.

[Editors' note: further clarifications were requested prior to acceptance, as described below.]

Thank you for resubmitting your work entitled “Active Invasion of Bacteria into Living Fungal Cells” for further consideration at *eLife.* Your revised article has been favorably evaluated by Richard Losick (Senior editor) and a member of the Board of Reviewing Editors. The manuscript has been improved but there are some remaining issues that need to be addressed before acceptance, as outlined below:

The manuscript still misses a clear statement of the authors that addresses their view of what's happening inside the cell. The schematic Figure 7 shows the initial phase of the infection process (clearly the main topic of the manuscript), but it still does not address the final stage of the symbiosis. The association of the bacteria with the fungal hyphae resembles the close interaction of photobionts with fungal hyphae in lichens (where the hydrophobin cell wall layer also covers the photobiont), but this association can not be described as “... in the cell.” If the authors are convinced that the bacteria are located in the cell, they should provide evidence for this or, if not, reformulate their strong statement.

---

## [Author Response]

*1) It is desirable that the authors provide causal evidence that it is indeed chitinase activity that is required for infection. This is of particular importance as the authors themselves stated that supernatants of bacterial cultures contained rather little chitinolytic activity (how much?)*. *The authors may thus consider answering any of the following questions: Is the bacterial chitinase identified indeed capable of loosening the cell wall of this fungus? Would chitinase mutants lacking enzymatic activity still be able to support symbiosis formation? Would chitinase inhibitors be able to block or slow down bacterial entry?*

Thank you for this comment. In this paper we have provided several lines of experimental evidence that the bacterial chitinase plays a key role in the bacterial invasion:

A) We have demonstrated the chitinase activity by heterologous expression of the symbiont's chitinase gene in E. coli and successfully performed an in vitro chitinase assay.

B) By 2D gel analyses we have shown that the chitinase is excreted; a mutant incapable of chitinase secretion is not able to penetrate the fungal cell wall.

C) Calcofluor staining visualized chitin in the Zygomycete cell wall.

D) Expression of the chitinase gene is upregulated in co-culture.

E) An engineered mutant of the endosymbiont lacking this chitinase gene is not able to penetrate the fungal cell wall. In contrast, this ability is retained in various other engineered mutants.

F) We observed different phenotypes (smooth/rough surface) of the fungus in the presence or absence of the symbiont's chitinase.

Please note that this set of results alone summarizes over 5 years of experiments. Because of the difficult handling of the symbionts and the incompatibility of methods established for related free-living bacteria, the construction of a single mutant usually takes several months. In this paper, we present five new mutants, amongst other results, that for the first time shed light on a general mechanism of bacterial entry into fungal hyphae. At this stage it is technically impossible to co-express mutated versions of chitinase, but I trust the reviewers do not believe that the chitinase gene product does not play a structural role, given the successful demonstration of chitinase activity by heterologous expression.

As to the quantification, the challenge is that chitinase activity is hardly detectable in the isolated wild type. We measured the activity of pure bacterial cultures and WT incubated with colloidal chitin to see it is response to only chitin (no response reg. gene expression), and the co-cultures. However, in the co-culture chitinase activities could also be attributed to fungal chitinases, and thus we would refer focusing on the gene expression data, which is more specific.

In order to demonstrate the loosening of the cell wall, we employed Calcofluor staining. Indeed we have shown that the chitinase mutant lacks the ability to form the endosymbiotic interaction.

Prior to our successful chitinase gene deletion experiments we had already tested various inhibitors such as azacytidin, peroxide, xanthine derivatives, e.g., caffeine, some metals, but the results were not conclusive. We refrained from pursuing such inhibitor experiments because it was not possible to obtain any commercially available Family 18 specific inhibitors, and it could not be guaranteed that they would be specific for the symbionts's chitinase. In addition, the bacterial symbionts are typically embedded in a matrix of exopolysaccharides, which also hampers mutant selection with antibiotics. Thus, the chances that chitinase inhibitors would provide clear-cut results are low. We are convinced that the functional analyses using the chitinase gene knockout described in this study provide a much clearer result.

*2) There is no mention on the mechanism of entry after cell wall lysis as bacteria still have to pass the fungal membrane. Given that in previous studies it was shown that bacteria are not encapsulated by a host membrane after entering the fungus, they should penetrate the membrane in a different way. Are there examples for that? Perhaps Co-staining of the membrane (FM4-64) and the bacteria (GFP) could help to solve the question. Perhaps there is endocytosis but the membrane dissolves quickly afterwards? The reference cited in the text (*[46]*) does, in my view, not rule out the possibility of endo/phagocytosis*.

The reviewers make an excellent point. Actually, we had previously tried to test the possibility of endocytosis by tracing the potential event with staining. The experiment proved to be more challenging than expected because the dye bound to fungus unequally and remained extracellular. After many attempts, we have now succeeded in staining the fungal membrane with styryl dye FM4-64 and the bacterial symbionts by bacteria-specific dye Syto 9. After one hour of co-incubation, we observed bacteria attached to the fungus but no fungal membrane surrounding the bacterium. After five hours, we detected bacteria within the hyphae, and again no fungal membrane was visible. We added micrographs to the paper as new Figure 7. Overall, endocytotic activity cannot be observed. We are not aware of any precedent for such a scenario, and we believe that elucidating the mechanism of dissolution/penetration of the hosts’ membrane alone would warrant a publication in a high-impact journal. Once again, we are grateful for the excellent suggestion and are happy that we could now add this nice result to the current manuscript.

The citation to the TEM images has been corrected (2007b/c).

*3) The manuscript requires revision in order to make it understandable for a broader readership. In particular, more precision is required regarding the description of the process at a cellular level and correct terms have to be used*.

*a) A schematic picture describing the (putative) infection process and the final stage of endosymbiosis would be very helpful for the reader. It would make it clear that the EM pictures provided by the authors address the penetration of the cell wall and not the uptake of the bacterium by the fungal cell. Unfortunately, the authors frequently mix up these terms and they should be more precise in that respect*.

The manuscript was carefully edited with this regard.

*b) The authors are not precise in the use of “mycelium” and “hyphae”*. *The bacteria infect the hyphae and not the mycelium. The authors state that “bacteria ... enter the fungal cell by fusing with their cell wall”. It is not clear what fusion with the cell wall means and how such a process could lead to entry of the fungal cell (the cell wall is extracellular!)*

We have added the schematic view of the process of local cell wall dissolution and invasion to the main body as Figure 7.

*c) In Figure 6–figure supplement 1 the authors show a model where chitinase is secreted by the T2SS. If this reviewer understood the manuscript correctly, chitinase is secreted by the classical secretion machinery and not by T2SS*.

Chitinases are secreted through type 2 secretion system which is also called general secretion system. Therefore the genes are abbreviated as “gsp” as well. An explanatory sentence has been integrated to the manuscript.

*[Editors' note: further clarifications were requested prior to acceptance, as described below*.*]*

*Thank you for resubmitting your work entitled “Active Invasion of Bacteria into Living Fungal Cells” for further consideration at eLife. Your revised article has been favorably evaluated by Richard Losick (Senior editor) and a member of the Board of Reviewing Editors*. *The manuscript has been improved but there are some remaining issues that need to be addressed before acceptance, as outlined below:*

*The manuscript still misses a clear statement of the authors that addresses their view of what's happening inside the cell. The schematic*
Figure 7
*shows the initial phase of the infection process (clearly the main topic of the manuscript), but it still does not address the final stage of the symbiosis. The association of the bacteria with the fungal hyphae resembles the close interaction of photobionts with fungal hyphae in lichens (where the hydrophobin cell wall layer also covers the photobiont), but this association can not be described as “... in the cell.” If the authors are convinced that the bacteria are located in the cell, they should provide evidence for this or, if not, reformulate their strong statement*.

In the revised version of the manuscript, we have addressed the ambiguous point regarding the location of the symbiotic bacteria. We added a sentence to the Introduction stating:

”The fungus harbors endosymbionts of the genus *Burkholderia*, which reside within the fungal cytosol, as shown by confocal laser scanning microscopy, transmission electron microscopy (EM) and freeze–fracture EM (44, 45, 47). The bacteria are harnessed by the fungus...”

We refer to the previously published papers addressing this issue.